# Zero-training Sentence Embedding via Orthogonal Basis

## Abstract

We propose a simple and robust non-parameterized approach for building sentence representations. Inspired by the Gram-Schmidt Process in geometric theory, we build an orthogonal basis of the subspace spanned by a word and its surrounding context in a sentence. We model the semantic meaning of a word in a sentence based on two aspects. One is its relatedness to the word vector subspace already spanned by its contextual words. The other is the word's novel semantic meaning which shall be introduced as a new basis vector perpendicular to this existing subspace. Following this motivation, we develop an innovative method based on orthogonal basis to combine pre-trained word embeddings into sentence representations. This approach requires zero training and zero parameters, along with efficient inference performance. We evaluate our approach on 11 downstream NLP tasks. Experimental results show that our model outperforms all existing zero-training alternatives in all the tasks and it is competitive to other approaches relying on either large amounts of labelled data or prolonged training time.

## 1 Introduction

The concept of word embeddings has been prevalent in NLP community in recent years, as they can characterize semantic similarity between any pair of words, achieving promising results in a large number of NLP tasks (Mikolov et al., 2013; Pennington et al., 2014; Salle et al., 2016). However, due to the hierarchical nature of human language, it is not sufficient to comprehend text solely based on isolated understanding of each word. This has prompted a recent rise in search for semantically robust embeddings for longer pieces of text, such as sentences and paragraphs.

Based on learning paradigms, the existing approaches to sentence embeddings can be categorized into two categories: i) parameterized methods and ii) non-parameterized methods.

**Parameterized sentence embeddings.** These models are parameterized and require training to optimize their parameters. SkipThought (Kiros et al., 2015) is an encoder-decoder model that predicts adjacent sentences. Pagliardini et al. (2018) proposes an unsupervised model, Sent2Vec, to learn an n-gram feature in a sentence to predict the center word from the surrounding context. Quick thoughts (QT) (Logeswaran & Lee, 2018) replaces the encoder with a classifier to predict context sentences from candidate sequences. Khodak et al. (2018) proposes $à\,la\,carte$ to learn a linear mapping to reconstruct the center word from its context. Conneau et al. (2017) generates the sentence encoder InferSent using Natural Language Inference (NLI) dataset. Universal Sentence Encoder (Yang et al., 2018; Cer et al., 2018) utilizes the transformer (Vaswani et al., 2017) for sentence embeddings. The model is first trained on large scale of unsupervised data from Wikipedia and forums, and then trained on the Stanford Natural Language Inference (SNLI) dataset. Wieting & Gimpel (2017) propose the gated recurrent averaging network (GRAN), which is trained on Paraphrase Database (PPDB) and English Wikipedia. Subramanian et al. (2018) leverages a multi-task learning framework to generate sentence embeddings. Wieting et al. (2015a) learns the paraphrastic sentence representations as the simple average of updated word embeddings.

**Non-parameterized sentence embedding.** Recent work (Arora et al., 2017) shows that, surprisingly, a weighted sum or transformation of word representations can outperform many sophisticated neural network structures in sentence embedding tasks. These methods are parameter-free and require no further training upon pre-trained word vectors. Arora et al. (2017) constructs a sentence embedding called SIF as a sum of pre-trained word embeddings, weighted by reverse document

frequency. Rücklé et al. (2018) concatenates different power mean word embeddings as a sentence vector in $p$-mean. As these methods do not have a parameterized model, they can be easily adapted to novel text domains with both fast inference speed and high-quality sentence embeddings. In view of this trend, our work aims to further advance the frontier of this group and make its new state-of-the-art.

In this paper, we propose a novel sentence embedding algorithm, Geometric Embedding (GEM), based entirely on the geometric structure of word embedding space. Given a $d$-dim word embedding matrix $\boldsymbol{A} \in \mathbb{R}^{d \times n}$ for a sentence with $n$ words, any linear combination of the sentence's word embeddings lies in the subspace spanned by the $n$ word vectors. We analyze the geometric structure of this subspace in $\mathbb{R}^d$. When we consider the words in a sentence one-by-one in order, each word may bring in a novel orthogonal basis to the existing subspace. This new basis can be considered as the new semantic meaning brought in by this word, while the length of projection in this direction can indicate the intensity of this new meaning. It follows that a word with a strong intensity should have a larger influence in the sentence's meaning. Thus, these intensities can be converted into weights to linearly combine all word embeddings to obtain the sentence embedding. In this paper, we theoretically frame the above approach in a QR factorization of the word embedding matrix $\boldsymbol{A}$. Furthermore, since the meaning and importance of a word largely depends on its close neighborhood, we propose the sliding-window QR factorization method to capture the context of a word and characterize its significance within the context.

In the last step, we adapt a similar approach as Arora et al. (2017) to remove top principal vectors before generating the final sentence embedding. This step is to ensure commonly shared background components, e.g. stop words, do not bias sentence similarity comparison. As we build a new orthogonal basis for each sentence, we propose to have disparate background components for each sentence. This motivates us to put forward a sentence-specific principal vector removal method, leading to better empirical results.

We evaluate our algorithm on 11 NLP tasks. In all of these tasks, our algorithm outperforms all non-parameterized methods and many parameterized approaches. For example, compared to SIF (Arora et al., 2017), the performance is boosted by 5.5% on STS benchmark dataset, and by 2.5% on SST dataset. Plus, the running time of our model compares favorably with existing models.

The rest of this paper is organized as following. In Section 2, we describe our sentence embedding algorithm GEM. We evaluate our model on various tasks in Section 3 and Section 4. Finally, we summarize our work in Section 5.

## 2 APPROACH

### 2.1 QUANTIFY NEW SEMANTIC MEANING

Let us consider the idea of word embeddings (Mikolov et al., 2013), where a word $w_i$ is projected as a vector $\boldsymbol{v}_{w_i} \in \mathbb{R}^d$. Any sequence of words can be viewed as a subspace in $\mathbb{R}^d$ spanned by its word vectors. Before the appearance of the $i$th word, $\boldsymbol{S}$ is a subspace in $\mathbb{R}^d$ spanned by $\{\boldsymbol{v}_{w_1}, \boldsymbol{v}_{w_2}, ..., \boldsymbol{v}_{w_{i-1}}\}$. Its orthonormal basis is $\{\boldsymbol{q}_1, \boldsymbol{q}_2, ..., \boldsymbol{q}_{i-1}\}$. The embedding $\boldsymbol{v}_{w_i}$ of the $i$th word $w_i$ can be decomposed into

$$
\begin{aligned}
\boldsymbol{v}_{w_i} &= \sum_{j=1}^{i-1} r_j \boldsymbol{q}_j + r_i \boldsymbol{q}_i \\
r_j &= \boldsymbol{q}_j^T \boldsymbol{v}_{w_i} \\
r_i &= \|\boldsymbol{v}_{w_i} - \sum_{j=1}^{i-1} r_j \boldsymbol{q}_j\|_2
\end{aligned}
\tag{1}
$$

where $\sum_{j=1}^{i-1} r_j \boldsymbol{q}_j$ is the part in $\boldsymbol{v}_{w_i}$ that resides in subspace $\boldsymbol{S}$, and $\boldsymbol{q}_i$ is orthogonal to $\boldsymbol{S}$ and is to be added to $\boldsymbol{S}$. The above algorithm is also known as **Gram-Schmidt Process**. In the case of rank deficiency, i.e., $\boldsymbol{v}_{w_i}$ is already a linear combination of $\{\boldsymbol{q}_1, \boldsymbol{q}_2, ...\boldsymbol{q}_{i-1}\}$, $\boldsymbol{q}_i$ is a zero vector and $r_i = 0$. In matrix form, this process is also known as *QR factorization*, defined as follows.

**QR factorization.** Define an embedding matrix of $n$ words as $\boldsymbol{A} = [\boldsymbol{A}_{:,1}, \boldsymbol{A}_{:,2}, ..., \boldsymbol{A}_{:,n}] \in \mathbb{R}^{d \times n}$, where $\boldsymbol{A}_{:,i}$ is the embedding of the $i$th word $w_i$ in a word sequence $(w_1, \ldots, w_i, \ldots, w_n)$. $\boldsymbol{A} \in \mathbb{R}^{d \times n}$ can be factorized into $\boldsymbol{A} = \boldsymbol{Q}\boldsymbol{R}$, where the non-zero columns in $\boldsymbol{Q} \in \mathbb{R}^{d \times n}$ are the orthonormal basis, and $\boldsymbol{R} \in \mathbb{R}^{n \times n}$ is an upper triangular matrix.

The process above computes the novel semantic meaning of a word w.r.t all preceding words. As the meaning of a word influences and is influenced by its close neighbors, we now calculate the novel orthogonal basis vector $\boldsymbol{q}_i$ of each word $w_i$ in its neighborhood, rather than only w.r.t the preceding words.

**Definition 1 (Contextual Window Matrix)** *Given a word $w_i$, and its $m$-neighborhood window inside the sentence $(w_{i-m}, \ldots, w_{i-1}, w_i, w_{i+1}, \ldots, w_{i+m})$ , define the contextual window matrix of word $w_i$ as:*

$$\boldsymbol{S}^i = [\boldsymbol{v}_{w_{i-m}}, \ldots, \boldsymbol{v}_{w_{i-1}}, \boldsymbol{v}_{w_{i+1}}, \ldots, \boldsymbol{v}_{w_{i+m}}, \boldsymbol{v}_{w_i}] \in \mathbb{R}^{d \times (2m+1)} \tag{2}$$

Here we shuffle $\boldsymbol{v}_{w_i}$ to the end of $\boldsymbol{S}^i$ to compute its novel semantic information compared with its context. Now the QR factorization of $\boldsymbol{S}^i$ is

$$\boldsymbol{S}^i = \boldsymbol{Q}^i \boldsymbol{R}^i \tag{3}$$

Note that $\boldsymbol{q}_i$ is the last column of $\boldsymbol{Q}^i$, which is also the new orthogonal basis vector to this contextual window matrix.

Next, in order to generate the embedding for a sentence, we will assign a weight to each of its words. This weight should characterize how much new and important information a word brings to the sentence. The previous process yields the orthogonal basis vector $\boldsymbol{q}_i$. We propose that $\boldsymbol{q}_i$ represents the novel semantic meaning brought by word $w_i$. We will now discuss how to quantify i) the novelty of $\boldsymbol{q}_i$ to other meanings in $w_i$, ii) the significance of $\boldsymbol{q}_i$ to its context, and iii) the corpus-wise uniqueness of $\boldsymbol{q}_i$ w.r.t the whole corpus.

## 2.2 NOVELTY

We propose that a word $w_i$ is more important to a sentence if its novel orthogonal basis vector $\boldsymbol{q}_i$ is a large component in $\boldsymbol{v}_{w_i}$. This can be quantified as a novelty score:

$$\alpha_n = \exp(\boldsymbol{r}_{-1} / \|\boldsymbol{v}_{w_i}\|_2) = \exp(\boldsymbol{r}_{-1} / \|\boldsymbol{r}\|_2) \tag{4}$$

where $\boldsymbol{r}$ is the last column of $\boldsymbol{R}^i$, and $\boldsymbol{r}_{-1}$ is the last element of $\boldsymbol{r}$.

**Connection to least square.** From QR factorization theory, the novel orthogonal basis $\boldsymbol{q}_i$ is also the normalized residual in the least square problem $\min \|\boldsymbol{C}\boldsymbol{x} - \boldsymbol{v}_{w_i}\|_2$, i.e. $\boldsymbol{q}_i^T \boldsymbol{v}_{w_i} = \boldsymbol{r}_{-1} = \min \|\boldsymbol{C}\boldsymbol{x} - \boldsymbol{v}_{w_i}\|_2$, where $\boldsymbol{C} = \boldsymbol{S}^i_{:,1:2m}$. And $\boldsymbol{q}_i^T \boldsymbol{v}_{w_i} = \boldsymbol{r}_{-1}$ is the minimum distance from word vector $\boldsymbol{v}_{w_i}$ to the hyper-plane spanned by $w_i$'s context words $w_{i-m}, \ldots, w_{i-1}, w_{i+1}, \ldots, w_{i+m}$.

It follows that $\alpha_n$ is the exponential of the normalized distance between $\boldsymbol{v}_{w_i}$ and the subspace spanned by its context.

## 2.3 SIGNIFICANCE

The significance of a word is related to how semantically aligned it is to the meaning of its context. To identify principal directions, i.e. meanings, in the contextual window matrix $\boldsymbol{S}^i$, we employ *Singular Value Decomposition*.

**Singular Value Decomposition.** Given a matrix $\boldsymbol{A} \in \mathbb{R}^{d \times n}$, there exists $\boldsymbol{U} \in \mathbb{R}^{d \times n}$ with orthogonal columns, diagonal matrix $\boldsymbol{\Sigma} = \mathrm{diag}(\sigma_1, ..., \sigma_n)$, $\sigma_1 \geq \sigma_2 \geq ... \geq \sigma_n \geq 0$, and orthogonal matrix $\boldsymbol{V} \in \mathbb{R}^{n \times n}$, such that $\boldsymbol{A} = \boldsymbol{U}\boldsymbol{\Sigma}\boldsymbol{V}^T$.

The columns of $\boldsymbol{U}$, $\{\boldsymbol{U}_{:,j}\}_{j=1}^n$, are an orthonormal basis of $\boldsymbol{A}$'s columns subspace and we propose that they represent a set of semantic meanings from the context. Their corresponding singular values $\{\sigma_j\}_{j=1}^n$, denoted by $\sigma(\boldsymbol{A})$, represent the importance associated with $\{\boldsymbol{U}_{:,j}\}_{j=1}^n$. The SVD of $w_i$'s contextual window matrix is $\boldsymbol{S}^i = \boldsymbol{U}^i \boldsymbol{\Sigma}^i \boldsymbol{V}^{iT} \in \mathbb{R}^{d \times (2m+1)}$. It follows that $\boldsymbol{q}_i^T \boldsymbol{U}^i$ is the coordinate of $\boldsymbol{q}_i$ in the basis of $\{\boldsymbol{U}^i_{:,j}\}_{j=1}^{2m+1}$.

Intuitively, a word is more important if its novel semantic meaning has a better alignment with more principal meanings in its contextual window. This can be quantified as $\|\sigma(\boldsymbol{S}^i) \odot (\boldsymbol{q}_i^T \boldsymbol{U}^i)\|_2$, where $\odot$ denotes element-wise product. Therefore, we define the significance of $w_i$ in its context to be:

$$\alpha_s = \frac{\|\sigma(\boldsymbol{S}^i) \odot (\boldsymbol{q}_i^T \boldsymbol{U}^i)\|_2}{2m + 1} \tag{5}$$

It turns out $\alpha_s$ can be rewritten as

$$\alpha_s = \frac{\|\boldsymbol{q}_i^T \boldsymbol{U}^i \boldsymbol{\Sigma}^i\|_2}{2m + 1} = \frac{\|\boldsymbol{q}_i^T \boldsymbol{U}^i \boldsymbol{\Sigma}^i \boldsymbol{V}^i\|_2}{2m + 1} = \frac{\|\boldsymbol{q}_i^T \boldsymbol{S}^i\|_2}{2m + 1} = \frac{\boldsymbol{q}_i^T \boldsymbol{v}_{w_i}}{2m + 1} = \frac{\boldsymbol{r}_{-1}}{2m + 1} \tag{6}$$

We use the fact that $\boldsymbol{V}^i$ is an orthogonal matrix and $\boldsymbol{q}_i$ is orthogonal to all but the last column of $\boldsymbol{S}^i$, $\boldsymbol{v}_{w_i}$. Therefore, $\alpha_s$ is essentially the distance between $w_i$ and the context hyper-plane, normalized by the context size.

## 2.4 CORPUS-WISE UNIQUENESS

Similar to the idea of inverse document frequency (IDF) (Sparck Jones, 1972), a word that is commonly present in the corpus is likely to be a stop word, thus its corpus-wise uniqueness is small. In our solution, we compute the principal directions of the corpus and then measure their alignment with the novel orthogonal basis vector $\boldsymbol{q}_i$. If there is a high alignment, $w_i$ will be assigned a relatively low corpus-wise uniqueness score, and vice versa.

### 2.4.1 COMPUTE PRINCIPAL DIRECTIONS OF CORPUS

As proposed in Arora et al. (2017), given a corpus containing a set of sentences, each sentence embedding is first computed as a linear combination of its word embeddings, thus generating a sentence embedding matrix $\boldsymbol{X} = [\boldsymbol{c}_1, \boldsymbol{c}_2, \ldots, \boldsymbol{c}_N] \in \mathbb{R}^{d \times N}$ for a corpus $\mathcal{S}$ with $N$ sentences. Then principal vectors of $\boldsymbol{X}$ are computed.

In comparison, we do not form the sentence embedding matrix $\boldsymbol{X}$ after we finalize the sentence embedding. Instead, we obtain an intermediate coarse-grained sentence embedding matrix $\boldsymbol{X}^c = [\boldsymbol{g}_1, \ldots, \boldsymbol{g}_N]$ as follows. Suppose the SVD of the sentence matrix of the $i$th sentence is $\boldsymbol{S} = [\boldsymbol{v}_{w_1}, \ldots, \boldsymbol{v}_{w_n}] = \boldsymbol{U} \boldsymbol{\Sigma} \boldsymbol{V}^T$. Then the coarse-grained embedding for the $i$th sentence is defined as:

$$\boldsymbol{g}_i = \sum_{j=1}^{n} f(\sigma_j) \boldsymbol{U}_{:,j} \tag{7}$$

where $f(\sigma_j)$ is a monotonically increasing function. We then compute the top $K$ principal vectors $\{\boldsymbol{d}_1, \ldots, \boldsymbol{d}_K\}$ of $\boldsymbol{X}^c$, with singular values $\sigma_1 \geq \sigma_2 \geq \ldots \geq \sigma_K$.

### 2.4.2 UNIQUENESS SCORE

In contrast to Arora et al. (2017), we select different principal vectors of $\boldsymbol{X}^c$ for each sentence, as different sentences may have disparate alignments with the corpus. For each sentence, $\{\boldsymbol{d}_1, \ldots, \boldsymbol{d}_K\}$ are re-ranked in descending order of their correlation with sentence matrix $\boldsymbol{S}$. The correlation is defined as $o_i = \sigma_i \|\boldsymbol{S}^T \boldsymbol{d}_i\|_2, 1 \leq i \leq K$. Next, the top $h$ principal vectors after re-ranking based on $o_i$ are selected: $\boldsymbol{D} = \{\boldsymbol{d}_{t_1}, \ldots, \boldsymbol{d}_{t_h}\}$, with $o_{t_1} \geq o_{t_2} \geq \ldots \geq o_{t_h}$ and their singular values in $\boldsymbol{X}^c$ are $\boldsymbol{\sigma}_d = [\sigma_{t_1}, \ldots, \sigma_{t_h}] \in \mathbb{R}^h$.

Finally, a word $w_i$ with new semantic meaning vector $\boldsymbol{q}_i$ in this sentence will be assigned a corpus-wise uniqueness score:

$$\alpha_u = \exp\left(-\|\boldsymbol{\sigma}_d \odot (\boldsymbol{q}_i^T \boldsymbol{D})\|_2 / h\right) \tag{8}$$

This ensures that common stop words will have their effect diminished since their embeddings are closely aligned with the corpus' principal directions.

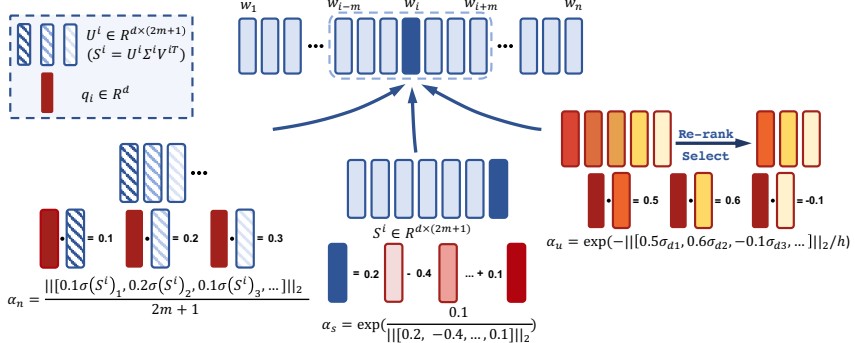

Figure 1: An illustration of GEM algorithm. Top middle: The sentence to encode, with words $w_1$ to $w_n$. And the contextual window of word $w_i$ is inside the dashed frame. Bottom middle: Form $\boldsymbol{S}^i$ for $w^i$, compute $\boldsymbol{q}_i$ and novelty score $\alpha_n$ (Section 2.1 and Section 2.2). Bottom left: Compute the SVD of $\boldsymbol{S}^i$ and significance score $\alpha_s$ (Section 2.3). Bottom right: Re-rank and select from principal components (orange blocks) and compute uniqueness score $\alpha_u$ (Section 2.4).

## 2.5 SENTENCE VECTOR

A sentence vector $\boldsymbol{c}_s$ is computed as a weighted sum of its word embeddings, where the weights come from three scores: a novelty score ($\alpha_n$), a significance score ($\alpha_s$) and a corpus-wise uniqueness score ($\alpha_u$).

$$\alpha_i = \alpha_n + \alpha_s + \alpha_u$$
$$\boldsymbol{c}_s = \sum_i \alpha_i \boldsymbol{v}_{w_i} \tag{9}$$

We provide a theoretical explanation of Equation (9) in Appendix.

**Sentence-Dependent Removal of Principal Components.** Arora et al. (2017) shows that given a set of sentence vectors, removing projections onto the principal components of the spanned subspace can significantly enhance the performance on semantic similarity task. However, as each sentence may have a different semantic meaning, it could be sub-optimal to remove the same set of principal components from all sentences.

Therefore, we propose the *sentence-dependent principal component removal* (SDR), where we re-rank top principal vectors based on correlation with each sentence. Using the method from Section 2.4.2, we obtain $\boldsymbol{D} = \{\boldsymbol{d}_{t_1}, ..., \boldsymbol{d}_{t_r}\}$ for a sentence $s$. The final embedding of this sentence is then computed as:

$$\boldsymbol{c}_s \leftarrow \boldsymbol{c}_s - \sum_{j=1}^{r} (\boldsymbol{d}_{t_j}^T \boldsymbol{c}_s) \boldsymbol{d}_{t_j} \tag{10}$$

Ablation experiments show that sentence-dependent principal component removal can achieve better result. The complete algorithm is summarized in Algorithm 1 with an illustration in Figure 1.

## 3 EXPERIMENTS

### 3.1 SEMANTIC SIMILARITY TASKS: STS BENCHMARK

We evaluate our model on the STS Benchmark (Cer et al., 2017), a sentence-level semantic similarity dataset from SemEval and SEM STS. The goal for a model is to predict a similarity score of two sentences given a sentence pair. The evaluation is by the Pearson's coefficient $r$ between human-labeled similarity (0 - 5 points) and predictions.

**Experimental settings.** We report two versions of our model, one only using GloVe word vectors (GEM + GloVe), and the other using word vectors concatenated from LexVec, fastText and PSL (Wieting et al., 2015b) (GEM + L.F.P). The final similarity score is computed as an inner product of

---

**Algorithm 1** Geometric Embedding (GEM)

---

1: **Inputs:**
    A set of sentences $\mathcal{S}$, vocabulary $\mathcal{V}$, word embeddings $\{\boldsymbol{v}_w \in \mathbb{R}^d \mid w \in \mathcal{V}\}$
2: **Outputs:**
    Sentence embeddings $\{\boldsymbol{c}_s \in \mathbb{R}^d \mid s \in \mathcal{S}\}$
3: **for** $i$th sentence $s$ in $\mathcal{S}$ **do**
4:     Form matrix $\boldsymbol{S} \in \mathbb{R}^{d \times n}$, $\boldsymbol{S}_{:,j} = \boldsymbol{v}_{w_j}$ and $w_j$ is the $j$th word in $s$
5:     The SVD is $\boldsymbol{S} = \boldsymbol{U\Sigma V}^T$
6:     The $i$th column of the coarse-grained sentence embedding matrix $\boldsymbol{X}^c_{:,i}$ is $\boldsymbol{U}(\sigma(\boldsymbol{S}))^3$
7: **end for**
8: Take first $K$ singular vectors $\{\boldsymbol{d}_1, ..., \boldsymbol{d}_K\}$ and singular values $\sigma_1 \geq \sigma_2 \geq ... \geq \sigma_K$ of $\boldsymbol{X}^c$
9: **for** sentence $s$ in $\mathcal{S}$ **do**
10:     Re-rank $\{\boldsymbol{d}_1, ..., \boldsymbol{d}_K\}$ in descending order by $o_i = \sigma_i \|\boldsymbol{S}^T \boldsymbol{d}_i\|_2, 1 \leq i \leq K$.
11:     Select top $h$ principal vectors as $\boldsymbol{D} = [\boldsymbol{d}_{t_1}, ..., \boldsymbol{d}_{t_h}]$, with singular values $\boldsymbol{\sigma}_d = [\sigma_{t_1}, ...., \sigma_{t_h}]$.
12:     **for** word $w_i$ in $s$ **do**
13:         $\boldsymbol{S}^i = [\boldsymbol{v}_{w_{i-m}}, ..., \boldsymbol{v}_{w_{i-1}}, \boldsymbol{v}_{w_{i+1}}, ..., \boldsymbol{v}_{w_{i+m}}, \boldsymbol{v}_{w_i}]$ is the contextual window matrix of $w_i$.
14:         Do QR decomposition $\boldsymbol{S}^i = \boldsymbol{Q}^i \boldsymbol{R}^i$, let $\boldsymbol{q}_i$ and $\boldsymbol{r}$ denote the last column of $\boldsymbol{Q}^i$ and $\boldsymbol{R}^i$
15:         $\alpha_n = \exp(\boldsymbol{r}_{-1}/\|\boldsymbol{r}\|_2), \alpha_s = \boldsymbol{r}_{-1}/(2m+1), \alpha_u = \exp(-\|\boldsymbol{\sigma}_d \odot (\boldsymbol{q}_i^T \boldsymbol{D})\|_2/h)$
16:         $\alpha_i = \alpha_n + \alpha_s + \alpha_u$
17:     **end for**
18:     $\boldsymbol{c}_s = \sum_{\boldsymbol{v}_i \in \boldsymbol{s}} \alpha_i \boldsymbol{v}_{w_i}$
19:     Principal vectors removal: $\boldsymbol{c}_s \leftarrow \boldsymbol{c}_s - \boldsymbol{D}\boldsymbol{D}^T \boldsymbol{c}_s$
20: **end for**

---

| Non-parameterized models | dev | test |
|---|---|---|
| **GEM + L.F.P** | **82.1** | **77.5** |
| GEM + LexVec | 81.9 | 76.5 |
| SIF (Arora et al., 2017) | 80.1 | 72.0 |
| LexVec | 58.78 | 50.43 |
| L.F.P | 62.4 | 52.0 |
| word2vec skipgram | 70.0 | 56.5 |
| Glove | 52.4 | 40.6 |
| ELMo | 64.6 | 55.9 |
| Parameterized models | | |
| Reddit + SNLI (Yang et al., 2018) | 81.4 | **78.2** |
| GRAN (Wieting & Gimpel, 2017) | **81.8** | 76.4 |
| InferSent (Conneau et al., 2017) | 80.1 | 75.8 |
| Sent2Vec (Pagliardini et al., 2018) | 78.7 | 75.5 |
| Paragram-Phrase (Wieting et al., 2015a) | 73.9 | 73.2 |

Table 1: Pearson's $r \times$ 100 on STSB

| | |
|---|---|
| **GEM + L.F.P** | 48.97 |
| Reddit + SNLI tuned | 47.44 |
| KeLP-contrastive1 | **49.00** |
| SimBow-contrastive2 | 47.87 |
| SimBow-primary | 47.22 |

Table 2: MAP on CQA subtask B

normalized sentence vectors. Since our model is non-parameterized, it does not utilize any information from the dev set when evaluating on the test set and vice versa. Hyper-parameters are chosen at $m = 7$, $h = 17$, $K = 45$, and $t = 3$ by conducing hyper-parameters search on dev set. Results on the dev and test set are reported in Table 1. As shown, on the test set, our model has a $5.5\%$ higher score compared with another non-parameterized model SIF, and $25.5\%$ higher than the baseline of averaging L.F.P word vectors. It also outperforms most parameterized models including GRAN, InferSent, and Sent2Vec. Of all evaluated models, our model only ranks second to Reddit + SNLI, which is trained on the Reddit conversations dataset (600 million sentence pairs) and SNLI (570k sentence pairs). In comparison, our proposed method requires no external data and no training.

| Model | Dim | Training time (h) | MR | CR | SUBJ | MPQA | SST | TREC | MRPC | SICK-R | SICK-E |
|---|---|---|---|---|---|---|---|---|---|---|---|
| *Non-parameterized models* | | | | | | | | | | | |
| **GEM + L.F.P** | 900 | 0 | **79.8** | **82.5** | **93.8** | **89.9** | **84.7** | **91.4** | **75.4/82.9** | **86.5** | **86.2** |
| GEM + GloVe | 300 | 0 | 78.8 | 81.1 | 93.1 | 89.4 | 83.6 | 88.6 | 73.4/82.3 | 86.3 | 85.3 |
| SIF | 300 | 0 | 77.3 | 78.6 | 90.5 | 87.0 | 82.2 | 78.0 | - | 86.0 | 84.6 |
| p-mean | 3600 | 0 | 78.4 | 80.4 | 93.1 | 88.9 | 83.0 | 90.6 | - | - | - |
| GloVe BOW | 300 | 0 | 78.7 | 78.5 | 91.6 | 87.6 | 79.8 | 83.6 | 72.1/80.9 | 80.0 | 78.6 |
| *Paraemterized models* | | | | | | | | | | | |
| InferSent | 4096 | 24 | 81.1 | 86.3 | 92.4 | 90.2 | 84.6 | 88.2 | 76.2/83.1 | 88.4 | 86.3 |
| Sent2Vec | 700 | 6.5 | 75.8 | 80.3 | 91.1 | 85.9 | - | 86.4 | 72.5/80.8 | - | - |
| SkipThought-LN | 4800 | 336 | 79.4 | 83.1 | 93.7 | 89.3 | 82.9 | 88.4 | - | 85.8 | 79.5 |
| FastSent | 300 | 2 | 70.8 | 78.4 | 88.7 | 80.6 | - | 76.8 | 72.2/80.3 | - | - |
| *à la carte* | 4800 | N/A | 81.8 | 84.3 | 93.8 | 87.6 | 86.7 | 89.0 | - | - | - |
| SDAE | 2400 | 192 | 74.6 | 78.0 | 90.8 | 86.9 | - | 78.4 | 73.7/80.7 | - | - |
| QT | 4800 | 28 | 82.4 | 86.0 | **94.8** | 90.2 | **87.6** | 92.4 | 76.9/84.0 | 87.4 | - |
| STN | 4096 | 168 | **82.5** | **87.7** | 94.0 | **90.9** | 83.2 | 93.0 | **78.6/84.4** | **88.8** | **87.8** |
| USE | 512 | N/A | 81.36 | 86.08 | 93.66 | 87.14 | 86.24 | **96.60** | - | - | - |

Table 3: Results on supervised tasks. Sentence embeddings are fixed for downstream supervised tasks. Best results for each task are underlined, best results from models in the same category are in bold. SIF results are extracted from Arora et al. (2017) and Rücklé et al. (2018), and some training time is collected from Logeswaran & Lee (2018).

## 3.2 SEMANTIC SIMILARITY TASKS: CQA

We evaluate our model on subtask B of the SemEval Community Question Answering (CQA) task, another semantic similarity dataset. Given an original question $Q_o$ and a set of the first ten related questions $(Q_1, ..., Q_{10})$ retrieved by a search engine, the model is expected to re-rank the related questions according to their similarity with respect to the original question. Each retrieved question $Q_i$ is labelled "PerfectMatch", "Relevant" or "Irrelevant", with respect to $Q_o$. Mean average precision (MAP) is used as the evaluation measure.

We encode each question text into a unit vector $\boldsymbol{u}$. Retrieved questions $\{Q_i\}_{i=1}^{10}$ are ranked according to their cosine similarity with $Q_o$. Results are shown in Table 2. For comparison, we include results from the best models in 2017 competition: SimBow (Charlet & Damnati, 2017), KeLP (Filice et al., 2017), and Reddit + SNLI tuned. Note that all three benchmark models require training, and SimBow and KeLP leverage optional features including usage of comments and user profiles. In comparison, our model only uses the question text without any training. Our model clearly outperforms both Reddit + SNLI tuned and SimBow-primary, and on par with KeLP model.

## 3.3 SUPERVISED TASKS

We further test our model on nine supervised tasks, including seven classification tasks: movie review (MR) (Pang & Lee, 2005), Stanford Sentiment Treebank (SST) (Socher et al., 2013), question-type classification (TREC) (Voorhees & Dang, 2003), opinion polarity (MPQA) (Wiebe et al., 2005), product reviews (CR) (Hu & Liu, 2004), subjectivity/objectivity classification (SUBJ) (Pang & Lee, 2004) and paraphrase identification (MRPC) (Dolan et al., 2004). We also evaluate on SICK similarity (SICK-R), the SICK entailment (SICK-E) (Marelli et al., 2014). The sentence embeddings generated are fixed and only the downstream task-specific neural structure is learned. The four hyper-parameters are chosen the same as in STS benchmark experiment. Results are in Table 3.

GEM outperforms all non-parameterized sentence embedding models, including SIF, p-mean (Rücklé et al., 2018), and BOW on GloVe. It also compares favorably with most of parameterized models, including *à la carte* (Khodak et al., 2018), FastSent (Hill et al., 2016), InferSent, QT, Sent2Vec, SkipThought-LN (with layer normalization) (Kiros et al., 2015), SDAE (Hill et al., 2016), STN (Subramanian et al., 2018) and USE (Yang et al., 2018). Note that sentence representations generated by GEM have much smaller dimension compared to most of benchmark models, and the subsequent neural structure has fewer learnable parameters. The fact that GEM does well on several classification tasks (e.g. TREC and SUBJ) indicates that the proposed weight scheme is able to recognize important words in the sentence. Also, GEM's competitive performance on sentiment tasks shows that exploiting the geometric structures of two sentence subspaces is beneficial.

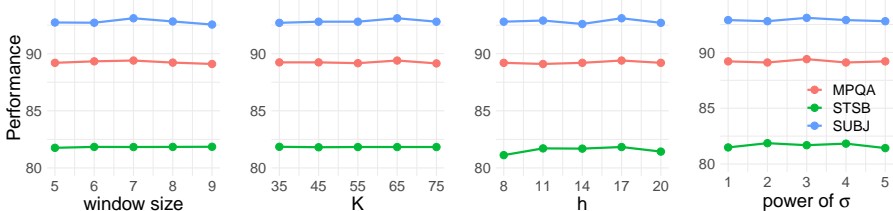

Figure 2: Sensitivity tests on four hyper-parameters, the window size $m$ in contextual window matrix, the number of candidate principal components $K$, the number of principal components to remove $h$, and the exponential power of singular value in coarse sentence embedding.

## 4 DISCUSSION

**Ablation Study.** As shown in in Table 4, every GEM weight $(\alpha_n, \alpha_s, \alpha_u)$ and proposed principal components removal methods contribute to the performance. As listed on the left, adding GEM weights improves the score by 8.6% on STS dataset compared with averaging three concatenated word vectors. The sentence-dependent principal component removal (SDR) proposed in GEM improves 0.3% compared to directly removing the top $h$ corpus principal components (SIR). Using GEM weights and SDR together yields an overall improvement of 19.7%. As shown on the right in Table 4, every weight contributes to the performance of our model. For example, three weights altogether improve the score in SUBJ task by 0.38% compared with only using $\alpha_n$.

| Configurations | STSB dev |
|---|---|
| Mean of L.F.P | 62.4 |
| GEM weights | 71.0 |
| GEM weights + SIR | 81.8 |
| GEM weights + SDR | 82.1 |

| Configurations | STSB dev | SUBJ |
|---|---|---|
| $\alpha_n$ + SDR | 81.6 | 93.42 |
| $\alpha_n, \alpha_s$ + SDR | 81.9 | 93.6 |
| $\alpha_n, \alpha_s, \alpha_u$ + SDR | 82.1 | 93.8 |

Table 4: Comparison of different configurations demonstrates the effectiveness of our model on STSB dev set and SUBJ. SDR stands for sentence-dependent principal component removal in Section 2.4.2. SIR stands for sentence-independent principal component removal, i.e. directly removing top $h$ corpus principal components from the sentence embedding.

**Sensitivity Study.** We evaluate the effect of all four hyper-parameters in our model: the window size $m$ in the contextual window matrix, the number of candidate principal components $K$, the number of principal components to remove $h$, and the power of the singular value in coarse sentence embedding, i.e. the power $t$ in $f(\sigma_j) = \sigma_j^t$ in Equation (7). We sweep the hyper-parameters and test on STSB dev set, SUBJ, and MPQA. Unspecified parameters are fixed at $m = 7$, $K = 45$, $h = 17$ and $t = 3$. As shown in Figure 2, our model is quite robust with respect to hyper-parameters.

**Inference speed.** We also compare the inference speed of our algorithm on the STSB test set with the benchmark models SkipThought and InferSent. SkipThought and InferSent are run on a NVIDIA Tesla P100, and our model is run on a CPU (Intel® Xeon® CPU E5-2690 v4 @2.60GHz). For fair comparison, batch size in InferSent and SkipThought is set to be 1. The results are shown in Table 5. It shows that without acceleration from GPU, our model is still faster than InferSent and is 54% faster than SkipThought.

| | GEM (CPU) | InferSent(GPU) | SkipThought (GPU) |
|---|---|---|---|
| Average running time (seconds) | **20.08** | 21.24 | 43.36 |
| Variance | 0.23 | 0.15 | 0.10 |

Table 5: Running time of GEM, InferSent and SkipThought on encoding sentences in STSB test set. GEM is run on CPU, InferSent and SkipThought is run on GPU. Data are collected from 5 trials.

## 5 CONCLUSIONS

We proposed a simple non-parameterized method [1] to generate sentence embeddings, based entirely on the geometric structure of the subspace spanned by word embeddings. Our sentence embedding evolves from the new orthogonal basis vector brought in by each word, which represents novel semantic meaning. The evaluation shows that our method not only sets up the new state-of-the-art of non-parameterized models but also performs competitively when compared with models requiring either large amount of training data or prolonged training time. In future work, we plan to consider multi-characters, i.e. subwords, into the model and explore other geometric structures in sentences.

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

# A   PROOF

The novelty score ($\alpha_n$), significance score ($\alpha_s$) and corpus-wise uniqueness score ($\alpha_u$) are larger when a word $w$ has relatively rare appearance in the corpus and can bring in new and important semantic meaning to the sentence.

Following the section 3 in Arora et al. (2017), we can use the probability of a word $w$ emitted from sentence $s$ in a dynamic process to explain eq. (9) and put this as following Theorem with its proof provided below.

**Theorem 1.** *Suppose the probability that word $w_i$ is emitted from sentence $s$ is[2]:*

$$\mathrm{p}[w_i|\boldsymbol{c}_s] \propto (\frac{\exp(\langle \boldsymbol{c}_s, \boldsymbol{v}_{w_i}\rangle)}{Z} + \exp(-(\alpha_n + \alpha_s + \alpha_u))) \tag{11}$$

*where $\boldsymbol{c}_s$ is the sentence embedding, $Z = \sum_{w_i \in \mathcal{V}} \exp(\langle \boldsymbol{c}_s, \boldsymbol{v}_{w_i}\rangle)$ and $\mathcal{V}$ denotes the vocabulary. Then when $Z$ is sufficiently large, the MLE for $\boldsymbol{c}_s$ is:*

$$\boldsymbol{c}_s \propto \sum_{w_i \in s} (\alpha_n + \alpha_s + \alpha_u)\boldsymbol{v}_{w_i} \tag{12}$$

**Proof:** According to Equation (11),

$$\mathrm{p}[w_i|\boldsymbol{c}_s] = \frac{1}{N}(\frac{\exp(\langle \boldsymbol{c}_s, \boldsymbol{v}_{w_i}\rangle)}{Z} + \exp(-(\alpha_n + \alpha_s + \alpha_u))) \tag{13}$$

Where $N$ and $Z$ are two partition functions defined as

$$N = 1 + \sum_{w_i \in \mathcal{V}} \exp(-(\alpha_n(w_i) + \alpha_s(w_i) + \alpha_u(w_i)))$$

$$Z = \sum_{w_i \in \mathcal{V}} \exp(\langle \boldsymbol{c}_s, \boldsymbol{v}_{w_i}\rangle) \tag{14}$$

The joint probability of sentence $s$ is then

$$p[s|\boldsymbol{c}_s] = \prod_{w_i \in s} p(w_i|\boldsymbol{c}_s) \tag{15}$$

To simplify the notation, let $\alpha = \alpha_n + \alpha_s + \alpha_u$. It follows that the log likelihood $f(w_i)$ of word $w_i$ emitted from sentence $s$ is given by

$$f_{w_i}(\boldsymbol{c}_s) = \log(\frac{\exp(\langle \boldsymbol{c}_s, \boldsymbol{v}_{w_i}\rangle)}{Z} + e^{-\alpha}) - \log(N) \tag{16}$$

$$\nabla f_{w_i}(\boldsymbol{c}_s) = \frac{\exp(\langle \boldsymbol{c}_s, \boldsymbol{v}_{w_i}\rangle)\boldsymbol{v}_{w_i}}{\exp(\langle \boldsymbol{c}_s, \boldsymbol{v}_{w_i}\rangle) + Ze^{-\alpha}} \tag{17}$$

By Taylor expansion, we have

$$f_{w_i}(\boldsymbol{c}_s) \approx f_{w_i}(0) + \nabla f_{w_i}(0)^T \boldsymbol{c}_s$$

$$= \text{constant} + \frac{\langle \boldsymbol{c}_s, \boldsymbol{v}_{w_i}\rangle}{Ze^{-\alpha} + 1} \tag{18}$$

Again by Taylor expansion on $Z$,

$$\frac{1}{Ze^{-\alpha} + 1} \approx \frac{1}{1 + Z} + \frac{Z}{(1 + Z)^2}\alpha$$

$$\approx \frac{Z}{(1 + Z)^2}\alpha \tag{19}$$

$$\approx \frac{1}{1 + Z}\alpha$$

---

[2]The first term is adapted from Arora et al. (2017), where words near the sentence vector $\boldsymbol{c}_s$ has higher probability to be generated. The second term is introduced so that words similar to the context in the sentence or close to common words in the corpus are also likely to occur.

The approximation is based on the assumption that $Z$ is sufficiently large. It follows that,

$$f_{w_i}(\boldsymbol{c}_s) \approx \text{constant} + \frac{\alpha}{1 + Z}\langle \boldsymbol{c}_s, \boldsymbol{v}_{w_i}\rangle \tag{20}$$

Then the maximum log likelihood estimation of $\boldsymbol{c}_s$ is:

$$\begin{aligned}
\boldsymbol{c}_s &\approx \sum_{w_i \in s} \frac{\alpha}{1 + Z}\boldsymbol{v}_{w_i} \\
&\propto \sum_{w_i \in s} (\alpha_n + \alpha_s + \alpha_u)\boldsymbol{v}_{w_i}
\end{aligned} \tag{21}$$

## B    EXPERIMENTAL SETTINGS

For all experiments, sentences are tokenized using the NLTK tokenizer (Bird et al., 2009) word-punct_tokenize, and all punctuation is skipped. $f(\sigma_j) = \sigma_j^t$ in Equation (7). In the STS benchmark dataset, our hyper-parameters are chosen by conducting parameters search on STSB dev set at $m = 7$, $h = 17$, $K = 45$, and $t = 3$. And we use the same values for all supervised tasks. The integer interval of parameters search are $m \in [5, 9]$, $h \in [8, 20]$, $L \in [35, 75]$ (at stride of 5), and $t \in [1, 5]$. In CQA dataset, $m$ and $h$ are changed to 6 and 15, the correlation term in section 2.4.2 is changed to $o_i = \|\boldsymbol{S}^T\boldsymbol{d}_i\|_2$ empirically. In supervised tasks, same as Arora et al. (2017), we do not perform principal components in supervised tasks.

## C    ENCODE A LONG SEQUENCE OF WORDS

We would like to give a clarification on encoding a long sequence of words, for example, a paragraph or a article. Specifically, the length $n$ of the sequence is larger than the dimension $d$ of pre-trained word vectors in this case. The only part in GEM relevant to the the length of the sequence $n$ is the coarse embedding in Equation (7). The SVD of the sentence matrix of the $i$th sentence is still $\boldsymbol{S} \in \mathbb{R}^{d \times n} = [\boldsymbol{v}_{w_1}, \ldots, \boldsymbol{v}_{w_n}] = \boldsymbol{U}\boldsymbol{\Sigma}\boldsymbol{V}^T$, where now $\boldsymbol{U} \in \mathbb{R}^{d \times d}$, $\boldsymbol{\Sigma} \in \mathbb{R}^{d \times n}$, and $\boldsymbol{V} \in R^{n \times n}$. Note that the $d + 1$th column to $n$th column in $\boldsymbol{\Sigma}$ are all zero. And Equation (7) becomes $\boldsymbol{g}_i = \sum_{j=1}^{d} f(\sigma_j)\boldsymbol{U}_{:,j}$. The rest of the algorithm works as usual.

