# OpenReview forum: "Zero-training Sentence Embedding via Orthogonal Basis"
_ICLR.cc/2019/Conference_

### Official Review · AnonReviewer1 · 2018-10-23
**Interesting idea with issues to resolve**

**Rating:** 5
**Confidence:** 4

**Review:**

This is a paper about sentence embedding based on orthogonal decomposition of the spanned space by word embeddings. Via Gram-Schmidt process, the sequence of words in a sentence is regarded as a sequence of incoming vectors to be orthogonalized. Each word is then assigned 3 scores: novelty score, significance score, and uniqueness score. Eventually, the sentence embedding is achieved as weighted average of word embeddings based on those scores. The authors conduct extensive experiments to demonstrate the performance of the proposed embedding. I think the idea of the paper is novel and inspiring. But there are several issues and possible areas to improve:

1. What if the length of the sentence is larger than the dimension of the word embedding? Some of the 3 scores will not be well-defined.

2. Gram-Schmidt process is sensitive to the order of the incoming vectors. A well-defined sentence embedding algorithm should not. I suggest the authors to evaluate whether this is an issue. For example, if by simply removing a non-important stop word at the begging of the sentence and then the sentence embedding changes drastically, then it indicates that the embedding is problematic.

3. I’m confused by the classification between training-free sentence embedding and unsupervised sentence embedding? Don’t both of them require training word2vec-type embedding?

4. The definition of the three scores seems reasonable, but requires further evidence to justify. For example, by the definition of the scores, do we have any proof that the value of \alpha indeed demonstrated the related importance level?

---

> ### Author Response · Authors · 2018-11-08
> **Response**
>
> Hi AnonReviewer1,
>
> Thanks for reviewing the paper and recognizing the novelty in our idea! Please find our response to the four points as follows.
>
> (1)
> In the case that the length of the sentence is larger than the dimension of the word embeddings, our algorithm still works fun. Sorry for the possible confusion and here are some clarifications:
> First, the novelty score and significance score are independent of the length of the sentence, so they are good.
>
> For the uniqueness score, the part that depends on the length of the sentence is the coarse embedding in eq(7). For the coarse embedding, now we have a sentence matrix S of size d*n, where d is word vector embedding size, n is the length of sentence, and n > d. The thin SVD of S is S = U*Sigma*V, where U is of size d*d, Sigma is of size d*n, and V is of size n*n. And the (d+1)th column to the nth column in Sigma is zeros, this is because S only has number of d singular values. In this case, the upper limit in eq(7) is n instead of d, and we have the coarse embedding from sentence matrix S. Therefore, in this case our model is fun. We’ll add explanation on this corner case in the appendix in the revised version (will submit very soon).
>
> (2)
> First, although we use Gram-Schmidt process (GS), GEM is not that sensitive to the order of words, explained as follows. For GS on n incoming vectors, if the last vector is fixed, the last orthogonal base vector computed is independent of the order of first n vectors. In our case, the word w_i is always shifted to the last column in the context window, and we only utilize the last orthogonal base vector, q_i, generated by GS. Therefore, no matter how the first (n-1) words in the context window are shifted, q_i is always the same. And those three scores stay the same for w_i.
>
> Second, as suggested in the review, we do some experiment of removing non-important stop words.
>
> S1: "The student is reading a physics book"
> S2: "student is reading a physics textbook"
> The cosine similarity between sentence vector of s1 and s2 given by GEM is 0.998
>
> sent1= "A man walks along walkway to the store"
> sent2= "man walks along walkway to the store"
> cosine similarity = 0.984
>
> sent1= "Someone is sitting on the blanket"
> sent2= "Someone is sitting on blanket"
> cosine similarity = 0.981
>
> The similarity scores are all very closed to 1, suggesting that sentence embeddings barely change.
>
> (3)
> We are sorry about the confusion. For “training-free”, we are trying to say that the sentence embedding model built upon word2vec-type embedding doesn’t require training and free of trained parameters, for example, SIF and GEM belongs to this training-free type. And “training-required” means the embeddings model needs training to update its parameters, for example skip-thoughs and InferSent. We plan to rename the two types as parameter-free and parameters-required in the revised version.
>
> (4)
> We do some experiments to show that the value of \alpha in GEM shows the relative importance level.
> First, assume that originally the GEM assigns a weight alpha_i for the word w_i,. On STS benchmark test set, GEM achieves 77.5 (Pearson’s r * 100). If now the weight is changed to 1/(alpha_i), the performance drops to 69.59. And the performance falls to 32.83, if the weight is exp(-alpha_i). These results show that if we are trying to assign small weight to words that GEM assign high alpha value, and then sentence embeddings performs very badly. This phenomenon indicates that the alpha value given by GEM reflects the related importance level.
>
> And we list some concrete examples of alpha value below:
>
> The sentence to encode: “there are two ducks swimming in the river”
> Alpha values (sorted) by GEM are: [ducks: 4.93198917, river:4.7221562 , swimming: 4.70170178, two: 3.87061874, are: 3.54588148, there: 3.23038324, in:3.045169, the:2.93566744]
> GEM assigns higher weight to informative words like “ducks” and “river”, and downplay stop words like “the” and “there”.
>
> The sentence to encode: “The stock market opens low on Friday”
> Alpha values (sorted) by GEM are: [lower: 4.94505258, stock: 4.93871886, closes: 4.78424269, market: 4.62267853, Friday: 4.51399687, the: 3.75456615, on: 3.70935467]
> GEM emphasizes informative words like “lower” and “closes”, and diminishes stop words like “the” and “there”.
>
> We sincerely look forward to your further feedbacks and evaluation.

---

### Official Review · AnonReviewer2 · 2018-11-02
**review of Zero-training Sentence Embedding via Orthogonal Basis**

**Rating:** 4
**Confidence:** 4

**Review:**

Paper overview: This paper proposes a new geometry-based method for sentence embedding from word embedding vectors, inspired by Arora et al (2017). The idea is to quantify the novelty,significance and corpus-wise uniqueness of each word. In order to do so, they analyze geometrically how the word vector of the target word relates to 1) the subspace created by the word-vectors in its context 2) its alignment with the meanings in its context (using SVD) 3) its presence in the all the corpus. For each of these aspects, they output a score or weight. The final sentence representation is a weighted average, using these scores, of the word vectors of the sentence.

Remarks and questions:
     1) In table 1, Glove and word2vec are word representations, how is the sentence representation computed here?
     2) The authors are not comparing to what is now considered the state of the art methods, such as Quick thoughts vectors (ICLR 2018, 'an efficient framework for learning sentence representations' by Logeswaran et al.), Transformer (Attention is all you need by Vaswani et al.) and ELMo (Deep contextualized word representations, by Peters et al.).


Points in favor:
    1) Results: The method gives the best performance for non-training methods with an +2 point improvement on average, although it cannot beat training methods (see Table 3, for instance).
   2) On the result tables, it should be reported also the std, not just the average, so the reader can evaluate if the difference between the methods is statistically significant.
    3) Inference speed: the method is fast (see table 5)
    4) stability of the results: The method is robust to slight changes in the hyperparameters such as the size of the window, number of principal components used, etc (see Fig 2)


Points against:
     The methods presented in the paper are not novel. The main novelties are the geometrical analysis on the contribution of each word of the sentence to the sentence overall semantic meaning, and the definition of the scores (eqs 4,6,8) that allow to improve the weighted average sentence representation (eq 9), an idea already present in Arora et al.'s paper.


 Conclusion:
     Although the geometric analysis of the paper is interesting, I dont think it is sufficient to justify a paper at ICLR, unless, after comparison with the other methods proposed previously, the proposed model is still competitive and the difference is statistically significant.

---

> ### Author Response · Authors · 2018-11-09
> **Response to AnonReviewer2**
>
> Hello AnonReviewer2,
>
> Thank you for the detailed and careful review. We appreciate your points in favor and against.
>
> About Remarks and questions:
> (1)
> For rows “Glove” and “word2vec” in table 1, the sentence embeddings are computed as the simple average of all word embeddings of words in the sentence.
>
> (2)
> Sorry if we didn’t make this clearer in the paper, but we’ve included results from Quick Thoughts and a very recent model using transformer in the first version of our paper. Quick Thoughts is denoted as “QT”, and their results are shown on table 3. “Reddit + SNLI” in table 1 and table 2 is a very recent and competitive transformer model, introduced in [1] and [2]. The model uses the transformer from “attention is all you need” as the encoder. And in the revised version, we include their results on supervised tasks in table 3, denoted as USE. We also include ELMo’s performance on STS benchmark in table 1. The sentence embeddings are computed as the mean of ELMo vector of each word.
>
> Besides, we did comparison with other very recent and even more competitive models published around mid 2018, for example “a lar carte” and STN in table 3.
>
> Comparison with these models mentioned above:
> On STSB dataset, GEM (77.5/82.1) clearly outperforms mean of ELMo (55.87/64.58), and is very close to the transformer model on test set (actually better than it on dev set). On supervised tasks, GEM’s performance is definitely better than some parameterized methods (like SkipThought, Sent2Vec and FastSent). And it’s still very competitive compared with parameterized SOTA models, for example, GEM is better than transformer model USE on SUBJ, MPQA, better than a lar carte on MPQA, TREC.
>
> About novelty:
> (1) We acknowledge that SIF is the first published work on using weighted sum of word vectors for sentence representation. And representing sentence as a composition (average, non-linear, p-mean etc.) of word vectors has been an active research topic before and after SIF (e.g. ref [3][4][5]). And we believe there are still much to explore on this direction.
>
> (2) On GEM’s novelty.
> Although our model utilizes the idea weighted sum of word vectors, GEM is significantly different from SIF, including following aspects:
> - To our knowledge, we are the first to adopt well-established numerical linear algebra to quantify the sentence semantic meaning and the importance of words. And this simple method proves to be competitive.
>
> - The weights in SIF depend on statistic of vocabularies on very large corpus (wikipedia). In contrast, the weights in GEM are directly computed from the sentence “on the scene”. Given a sentence and its context, GEM is ready to go, independent of prior statistical knowledge of words.
>
> - In GEM, the components in weights are all computed from numerical linear algebra. And SIF directly have a hyper-parameter term in the weights, i.e. the smooth term.
>
> - As suggested by experiments in table 1 and 3, GEM outperforms SIF by significant margin.
>
> Thanks for your time again. Hope that our response addresses your concern. We kindly ask for your further evaluation and opinions.
>
> Reference:
> [1] Cer, Daniel, et al. "Universal sentence encoder." arXiv preprint arXiv:1803.11175 (2018).
> [2] Yang, Yinfei, et al. "Learning Semantic Textual Similarity from Conversations." arXiv preprint arXiv:1804.07754 (2018).
> [3] Wieting, John, et al. "Towards universal paraphrastic sentence embeddings." arXiv preprint arXiv:1511.08198 (2015).
> [4] Wieting, John, and Kevin Gimpel. "Revisiting recurrent networks for paraphrastic sentence embeddings." arXiv preprint arXiv:1705.00364 (2017).
> [5] Rücklé, Andreas, et al. "Concatenated $ p $-mean Word Embeddings as Universal Cross-Lingual Sentence Representations." arXiv preprint arXiv:1803.01400 (2018).

---

### Official Review · AnonReviewer3 · 2018-11-06
**missing a lot of details in the proposed model**

**Rating:** 5
**Confidence:** 4

**Review:**

The paper presented a new training-free way of generating sentence embedding. The proposed work is along the same motivation from Arora et al.,  2017. A systematic analysis has been done on a number of tasks to show the strong performance (close or higher than the specifically "supervised" strategies).

- I suggest the author to re-ward the category terms of the existing methods. Un-supervised and training-free are confusing. Unsupervised and supervised should be all in a group of training-required methods. unsupervised in this paper is more task-agnostic but domain specific and supervised is to extract sentence emb that is prediction task specific.

- The evaluation tasks are rich but not clearly stated. For instance, the supervised taske are only discussed at high-level. Not clear what each task is and how one should interpret the results from each experiments.  The way author presented it suggests the detail here were not important. It is also good to include discussion on how the baseline algorithms are tuned and/or trained on these tasks. Readers cannot reproduce the same results based on the current paper.

- Notation and Math:
--r-1 in (4) is not clear as \mathbf{r} is not defined properly
--based on sec 2.2., it is easy to motivate the novelty score from subspace projection rather than QR/GS;
-- a_n and a_s are both functions of r_{-1} which is the perp. energy of the words w.r.t. its contexts. Is there a fundamental difference?
-- Figure 1 is a little bit confusing. Not clear what is word and what is a sentence/corpus.
-- in Eq(8), better not to use r as it confuses with the GS coeffs.
-- 2.4.1 is a bit confusing, sentence embeddings c_1, \ldot, c_N are introduced, but so far no sentence embedding has been formally introduced. Is this initialized from some heuristic? It is confusing in the sense that eq (9) c_s are defined by a_u, but a_u defined in eq (8) depends on sigma_d that relies on X^{c}, which is a funcion of all c_s's.
-- there are several parameters for GEM, please add some discussion on how these are selected in each of the evaluated tasks.

---

> ### Author Response · Authors · 2018-11-10
> **Response to AnonReviewer3**
>
> Hello AnonReviewer3,
>
> We appreciate your comprehensive review and questions. Please find our response below.
>
> (1) About re-word the categories. Thanks for your suggestion. In the revised version submitted, we categorize sentence embeddings methods into two types, one is non-parameterized methods, including GEM and SIF, that don’t depend on parameters or need training. The other type is parameterized methods, such as InferSent and QuickThoughts, that need supervised/unsupervised training to update the parameters.
>
> (2) About supervised tasks. We are sorry for the confusion. in section 3.3, we add a description of supervised tasks (first paragraph) and an analysis of results (the end of second paragraph).
>
> (3) On “how the baseline algorithms are tuned and/or trained on these tasks”.
> On the supervised tasks, the performance of baseline model “GloVe BOW” is extracted from ref[1]. On STSB dataset, results of baseline model “word2vec skipgram” and “Glove” are extracted from the official website of STSB dataset (http://ixa2.si.ehu.es/stswiki/index.php/STSbenchmark). “LexVec”, “L.F.P” and “ELMo” are from experiments run by us. As noted in the “experimental settings” section in the appendix, sentences are tokenized using the NLTK wordpunct tokenizer, and then all punctuation is then skipped. Sentence vectors are just mean of word representations, and the similarity score is the cosine similarity of two vectors.
>
> (4) About \mathbf{r}. In the line under Eq(4), we mention that \mathbf{r} is the last column of R^i. And R^i is defined in Eq(3).
>
> (5) About GS and subspace projections. We agree that subspaces projection is more mathematically concise compared with GS. The reason why we still use GS to introduce novelty score is that GEM is motivated by the fact when a sentence is formed, different words bring in different meaning to this sentence one by one, and GS is appropriate to describe this process by yielding the orthogonal basis vectors one by one.
>
> (6) Although a_n and a_s are both functions of r_{-1}, they describe different quantities. Note that a_s is initially computed as q_i’s alignment with the meanings in its context. And Eq(6) shows that a_s is r_{-1}, i.e. the l_2 norm of q_i, divided by a constant. a_s is trying to quantify the absolute significance/magnitude of the new semantic meaning q_i.
>
> In contrast, a_n is a function (exponential) of r_{-1} divided by l_2 norm of r, i.e. a function of the “proportion” of q_i in word w_i. Note that ||r||_2 = ||v_{w_i}||_2, and r_{-1} = ||q_i||_2. Therefore, a_n is quantifying that among all the information that w_i is trying to ship, what’s proportion of the new meaning q_i?
>
> (7) On fig 1. We apologize for the possible ambiguity. The sentence is represented by a sequence of blue block in the top middle, marked as w_1 … w_{i-m} … w_i … w_{i+m} … w_n. And we didn’t show the corpus in fig 1, and instead we show the top K principal vectors of X^c as those orange/yellow blocks on the right. And more descriptions are added to the caption of fig 1.
>
> (8) In eq(8), we change the notation “r” to “h”. Thanks for your suggestion.
>
> (9) On “2.4.1 is a bit confusing”.
> We think you referred to the matrix in the first paragraph in 2.4.1. The first paragraph is a revisit of the method in SIF. The formal desription of GEM starts from the second paragraph. We form a matrix X^c and its ith column is given by eq(7). Eq(7) is independent of a_u, a_n and a_s, and it’s computed using the singular values and singular vectors of the sentence matrix $\mS$. And then we use X^c and q_i to compute a_u.
>
> (10) In the STS benchmark dataset, our hyper-parameters are chosen by conducting parameters search on STSB dev set at m = 7, h = 17, K = 45, and t = 3. And we use the same values for all supervised tasks. The integer interval of parameters search are m ∈ [5, 9], h ∈ [8, 20], L ∈ [35, 75] (at stride of 5), and t ∈ [1, 5]. And we use the same values for all supervised tasks. We add the discussion to the “experimental settings” section in the appendix.
>
> Thanks for your time and we hope that our response has addressed your questions. Look forward to your suggestion and evaluation.
>
> Reference:
> [1] Conneau, Alexis, et al. "Supervised learning of universal sentence representations from natural language inference data." arXiv preprint arXiv:1705.02364 (2017).

---

### Public Comment · (anonymous) · 2018-10-08
**Evaluation with uSIF?**

Very interesting paper! I was wondering how your method compared against uSIF (https://github.com/kawine/usif), a variant of SIF with no hyperparameter tuning. uSIF did much better than SIF on the STS tasks, so I'd be interested in seeing how it does against your method here.

---

> ### Author Response · Authors · 2018-10-08
> **Performance compared with uSIF**
>
> Thanks for your interest! uSIF is evaluated on SST(80.7), SICK-R(83.8), SICK-E(81.1) and STSB test(79.5) (http://aclweb.org/anthology/W18-3012 ). And our model achieves performance of 84.7, 86.5, 86.2 and 77.5 respectively. We will cite uSIF in the revised version.

---

> > ### Public Comment · (anonymous) · 2018-10-10
> > **Using the same architecture?**
> >
> > Thanks for your reply! For what it's worth, I've noticed that a lot of these sentence embedding papers use different architectures for supervised tasks. Some use really simple ones, and others more complex ones. For example, with uSIF, the architecture was relatively simple, which is probably why the SIF scores it reports were also lower than what was reported in the original SIF paper.
> >
> > If it's not too much work, I think it'd be worth trying your architecture on some of other embedding types (SIF, USIF, infersent, etc.) I think it'd give us a much better idea of how much of a difference the embeddings are making as opposed to the architecture.

---

> > > ### Author Response · Authors · 2018-10-10
> > > **Clarification on structure for supervised tasks**
> > >
> > > Hi,
> > >
> > > GEM uses pretty simple and standard structure for supervised tasks, just logistic regression with one hidden layer. In terms of complexity and structure, the architecture we used is the same as SIF and [1]. Most of the evaluations are implemented using standardized evaluation tool SentEval[2](will cite in the revised version).
> > >
> > > Reference:
> > > [1] Wieting, John, et al. "Towards universal paraphrastic sentence embeddings." arXiv preprint arXiv:1511.08198 (2015).
> > > [2] Conneau, Alexis, and Douwe Kiela. "SentEval: An Evaluation Toolkit for Universal Sentence Representations." arXiv preprint arXiv:1803.05449 (2018).

---

### Public Comment · (anonymous) · 2018-10-09
**Difference with Sentences as subspaces**

Hello!

Do you represent sentence as a subspace? How are the observations different from https://arxiv.org/abs/1704.05358? How is it different from SIF?

Cheers!

---

> ### Author Response · Authors · 2018-10-10
> **GEM compared with SIF and "Sentences as subspaces"**
>
> Thanks for your comment! Our model (GEM) is quite different from both SIF and "Sentences as subspaces" as follows.
>
> First, compared with SIF, GEM generates the weight for each word in a completely different way. In SIF, weight is a function of IDF and a hyperparameter. In GEM, weight is computed by capturing the new semantic meaning brought in by each word (section 2.2, 2.3, 2.4). What’s more, principal components removal method is different in SIF and GEM. GEM proposes sentence-dependent principal component removal (SDR, section 2.5), and principal components are generated from coarse-grained sentence embedding matrix (section 2.4.1). In contrast, SIF removes the very same components from each sentence.
>
> Second, we were aware of the paper “low-rank subspaces” paper mentioned in your comment. That paper developed a very interesting method to compare the similarity of a PAIR of sentences, by using the principal angles between two low-rank approximation sentence matrices. We'll cite it in the revised version. It's true that “Sentences as subspaces” and our methods both begin with writing sentences in a matrix form. However, GEM is about generating a sentence vector for each sentence, using the geometric properties only in the single sentence to decode, while the other one is about generating a similarity score for a sentence pair, by comparing two subspaces. And in GEM, the final representation of a sentence is a vector, not a subspace.

---

### Public Comment · (anonymous) · 2018-11-02
**Regarding the principal component removal**

Hi! Really interesting work.

If I understand correctly, the principal component computation takes place across sentence embeddings in the test set. So, in particular, for a downstream task like STSB, the matrix X^c, would have coarse grained embeddings of sentences in the test set right? (and then the principal components are calculated)

I am not saying that this is a good/bad way and I believe SIF also does the principal component computation in this manner. While this cleverly utilizes information across sentences in the test set, I guess this can be a problem when you are given just one sentence at a time and you want to compute its embedding.

So, would the method also work if just two query sentences are passed in and it has to measure the similarity between them?

Thanks a lot!

---

> ### Author Response · Authors · 2018-11-02
> **For the case of embedding only one sentence**
>
> Hello,
>
> Thanks for reading our paper and your question.
>
> 1) Yes, you are right. The matrix X^c has coarse grained embeddings of sentences in the test set.
>
> 2) For corner case case of embedding a set of one sentence, GEM still works out with some simple adjustments. Some possible adjustments include: first, one can have a "background" context corpus in the very beginning. For example, if you want to encode a sentence about politics, you can calculate the principal components on some political articles dataset, which is regarded as the corpus now. Also in research datasets, the training set can always serve as the corpus for the test set. Second, in real-life engineering, GEM can keep and update a cache of the coarse embeddings of history queries. And this cache can serve as the corpus. The corner case is not explicitly taken care of in the pseudo code in our paper (and it's the same for SIF paper).
>
> 3) If you pass in two sentences, the algorithm works fun as usual. Also as pointed in 2), one can always can have a cache of coarse embeddings generated from previous queries or simply have a background context corpus.
>
> Thank you.

---

> > ### Public Comment · (anonymous) · 2018-11-02
> > **Thanks for your prompt and detailed response**
> >
> > Yes, I agree that the expected thing GEM/SIF should do is to estimate the principal components on a background context corpus.
> >
> > - But, right now, I believe that the evaluation for GEM (as well as SIF) is at an advantage in comparison to other methods, which don't use information across sentences in the test set.
> >
> > - STSB has a training set (which can be the "background" context corpus), and I think the comparison would be more accurate and fair if the principal component were estimated on the basis of it.
> >
> > - This doesn't take anything against your particular method which might be well placed in a scenario where you have the background corpus, but I think it's a good idea for the community to understand this nuance and having fair comparisons.
> >
> > - Regarding two sentences: I might be wrong here, but doesn't the quality of estimated principal components diminish with number of data samples (two sentences versus the entire training set previously).
> >
> > Thanks a lot again.

---

### Meta-Review · Area_Chair1 · 2019-01-24
**Interesting idea but in need of more clarity**

**Recommendation:** Reject
**Confidence:** 4

**Metareview:**

The paper proposes a simple approach for computing a sentence embedding as a weighted combination of pre-trained word embeddings, which obtains nice results on a number of tasks.  The approach is described as training-free but does require computing principal components of word embedding subspaces on the test set (similarly to some earlier work).  The reviewers are generally in agreement that the approach is interesting, and the results are encouraging.  However, there is some concern about the clarity of the paper and in particular the placement of the work in relation to other methods.  There is also a bit of concern about whether there is sufficient novelty compared to Arora et al. 2017, which also compose sentence embeddings as weighted combinations of word embeddings, and also use a principal subspace of embeddings in the test set.  This AC feels that the method here is sufficiently different from Arora et al., but agrees with the reviewers that the paper clarity needs to be improved, so that the community can appreciate what is gained from the new aspects of the approach and what conclusions should be drawn from each experimental comparison.